# Transition Interventions for Adolescents on Antiretroviral Therapy on Transfer from Pediatric to Adult Healthcare: A Systematic Review

**DOI:** 10.3390/ijerph192214911

**Published:** 2022-11-12

**Authors:** Olubukola Esther Jegede, Brian van Wyk

**Affiliations:** School of Public Health, University of the Western Cape, Cape Town 7535, South Africa

**Keywords:** transition intervention, antiretroviral therapy, adherence, retention in care, viral suppression, adolescents

## Abstract

Globally, adolescents living with HIV (ALHIV) experience poor health outcomes such as low retention in care, ART non-adherence and viral non-suppression. These outcomes coincide with the period during and after their transition from pediatric to adult healthcare. This study aimed to systematically describe the compendium of transition interventions and synthesize the effects of such transition interventions on adherence to ART, retention in care and viral load suppression. Seven databases and Google Scholar were searched and the review findings were reported according to the Preferred Reporting Items Stipulated for Systematic Reviews and Meta-Analyses. The risk of bias and the strength of evidence were assessed using the National Institutes of Health quality assessment tool for observational cohort and cross-sectional studies. Seven studies (two cross-sectional, two retrospective cohort and three prospective cohort studies), with sample sizes ranging from 13 to 192, were included in the narrative synthesis. There was high-quality evidence that these interventions—*Individualized care plans, communication, psychological support*, and *health and sexual education* and *mHealth*—improved adherence, retention in care and viral load suppression at post-transition over the short and long term. In contrast, *group transition* intervention produced weak quality evidence. Hence, transition interventions including a combination of the high-quality evidenced interventions mentioned above can improve treatment outcomes for adolescents on ART.

## 1. Introduction

Poor health outcomes among ALHIV are a primary public health concern globally [1,2]. In 2020 alone, 32,000 adolescents died due to AIDS-related deaths [1]. It is worthy of note that these poor health outcomes are exacerbated during and after the ALHIV transfer from pediatric to adult healthcare [3,4,5]. Transition is defined as “*the purposeful, planned movement of adolescents and young adults with chronic physical and medical conditions from child/adolescent-centered to adult-oriented healthcare systems*” [6] (p.1). The care transition in adolescents is a critical period in their lives because it coincides with the period of physiological and psychological development [7] and adulthood habits establishment [8]. Adolescents move from the pampering settings of the pediatrics clinic to the self-dependent settings of the adult clinic [9,10]. Hence, a structured transition is widely considered to be vital for the continuity of care and persistent engagement in care among ALHIV. 

Transition policies exist in the literature such as the American Academy of Pediatrics (AAP) transition policy statement. This policy highlights the basic elements of transition such as transition protocol development, transition preparedness, strong connection between pediatric/adolescent clinic and adult clinic and continuous evaluation of implemented transition strategies [11]. However, implementing and integrating these elements remains a challenge in clinical settings [12]. For instance, transition readiness being a critical component of foundation building for transition preparedness has no gold standards with regard to its measuring tools [13]. In addition to the challenge of implementation and integration of transition policies high-income countries face, there is a lack of transition policies and protocols at national and clinical levels in most of Sub-Saharan Africa [9,14]. Hence, there is a need for evidence-based research to tackle these bottlenecks and improve treatment outcomes for ALHIV. 

Systematic reviews on the impact, effectiveness and treatment outcomes of transition interventions in adolescents with chronic diseases have been reported in the literature, but none of these studies focused on ALHIV [15,16,17]. No systematic review has been conducted to describe transition interventions for ALHIV or document the evidence of the effects and effectiveness of such interventions. This is a critical gap in the literature because later adolescents report the worst treatment outcomes on ART [9,18]. Systematically reviewing these interventions can provide robust information on the strength of evidence from which recommendations for policy and practice can be made [19,20]. Additionally, describing and synthesizing evidence on the effects of transition interventions targeted at ALHIV will explain what works and inform reliable best practices, such as developing sound protocols that can be implemented and bringing about better health outcomes for ALHIV. Hence, we conducted a review of studies that describe interventions to guide the transition of adolescents on antiretroviral therapy (ART) on transfer from pediatric to adult care and documented the evidence of their effects on adherence, retention in care and viral load suppression.

This paper reports on a systematic review of transition interventions to improve adherence, retention in care and viral load suppression, assessed the strength of evidence effectiveness and provide recommendations for policy, clinical practice and further research. 

## 2. Materials and Methods

The current study is registered with PROSPERO (“an international database of prospectively registered systematic reviews with health-related outcomes”) (REF: CRD42021273205). The study utilized a systematic review study design following the Preferred Reporting Items stipulate for the Systematic Reviews and Meta-Analyses (PRISMA) 2020 statement

### 2.1. Study Selection and Data Extraction

The review question—namely, What are the effects of transition interventions to improve health outcomes among adolescents living with HIV transitioning from pediatric to adult healthcare?—was formulated following PICOT (population, intervention, comparative, outcomes and time) criteria. These criteria are explained and presented in Table 1. 

Two independent reviewers (O.E.J. & B.V.W.) screened the title and abstracts of the 606 studies identified according to the PICOT criteria. Full texts of the eligible studies were retrieved and reviewed by the two reviewers using Rayyan software and the conflict of judgment between the two reviewers was resolved through discussion. Reasons for excluding studies at this stage were documented. The processes involved in the study selection were diagrammatically represented in a PRISMA flow chart (Figure 1).

A data extraction form was designed and piloted to gather information on the author, year of publication, country, study population (sample size), study design/outcome, intervention description, outcomes and results. In cases where additional information was needed, authors of such articles were contacted via email.

### 2.2. Quality Assessment and Risk of Bias

The National Institutes of Health (NIH) quality assessment tool for observational cohort and cross-sectional studies was used for this review [21]. The tool comprises fourteen questions used to detect flaws in study methods or implementation, sources of bias, confounding, study power, the strength of causality and other factors, and provide collated answers to the study’s overall quality as “Good”, “Fair”, or “Poor.” A “**Good**” quality rating implies a low risk of bias, a “**fair**” quality rating implies a moderate risk of bias and a “**poor**” quality rating implies a high risk of bias. [21]. Studies were rated poor quality if exposure and/or outcome measurements were not clearly defined and confounding variables were not adjusted statistically. 

### 2.3. Data Synthesis and Analysis

Quantitative data from included studies were collected using Microsoft Excel statistical package. Due to high heterogeneity in the included studies a meta-analysis was impossible, the review adopted a narrative synthesis approach. The narrative synthesis structurally summarized and described the characteristics of the studies: study population, intervention description and study design/outcome, findings, and quality [22]. 

## 3. Results

### 3.1. Identification of Relevant Studies

A PRISMA flow diagram (Figure 1) shows the diagrammatic presentation of the processes followed in achieving the seven included studies for narrative synthesis. We identified 1009 studies from the search of Seven databases (PubMed, Scopus, Web of Science, Ebscohost, CINAHL, Science Direct and the World Health Organization’s (WHO’s) library) and (Google scholar). Three additional studies were added manually by searching references of the included studies. In total, 1012 studies were identified. Deduplication led to the exclusion of 406 studies and 606 studies went through title and abstract screening. After this, 552 studies were excluded, while the remaining 54 studies went through full-text screening. After the full-text screening, 47 additional studies were excluded because they did not meet the eligibility criteria. Only seven studies met the eligibility criteria and were included in the narrative synthesis.

### 3.2. Study Characteristics

The characteristics of the included studies are summarized in Table 2. The year of publication of included studies ranged from 2014 to 2020. Most studies (*n* = 5) were conducted in high-income countries: two in Italy [23,24] and the United States of America [25,26] and one in Sweden [27]. The remaining two studies [28,29] were conducted in Thailand, an upper-middle-income country [30]. 

### 3.3. Description of Transition Interventions

The transition interventions can be classified as pre-, during and post-transition. One study focused solely on pre-transition, three focused on during-transition, and the remaining three incorporated pre-, during and post-transition interventions. In this review, no study focused solely on post-transition intervention. 

#### 3.3.1. Pre-Transition Intervention

Only one study focused solely on pre-transition intervention [28]. The intervention prepared older adolescents ages 14 to 22 years for the transition by exposing them to HIV and health-related knowledge, which can bring about positive health outcomes during and after their transition to adult healthcare. 

#### 3.3.2. During-Transition Intervention

Three studies focused solely on during-transition intervention [25,27,29]. The first study implemented a transition reception that was neither for pediatrics nor adults but for HIV-positive adolescents preparing to transition into adult care [27]. This reception runs every month, but the appointment is every three months per patient. At first, adolescents received care from the multidisciplinary staff at the reception and then later, they received care from both the reception personnel and the adult healthcare personnel. The transfer reception provided them with individualized care according to their need and allowed them to consult a sexologist regarding their concerns about their sexuality. 

The second study assessed a youth-focused transition intervention—Access Early Care (ACE) compared to the Standard of Care (SOC) [25]. The ACE served as a transition clinic for those with a high risk of attrition and viremia. They have been exposed to additional phone or text message bi-directional communication from their peer navigator and care from four physicians with combined internal medicine and pediatric training.

For the third study, the during-transition intervention consisted of interdepartmental case conferences that involved pediatric and adult clinic staff, hospital home care team and clinical psychologist consultants [29]. The adolescents were involved in a transition camp that focuses on antiretroviral management, HIV transmission and navigating adult HIV care settings. The study also involved a group transition system that ensures the adolescent cohort receives an initial appointment and an appointment to present at the adult clinic together. Coupled with this is the presence of a pediatric provider in their first few appointments at the adult clinic to ensure proper integration through the provision of immediate support and answers to questions. 

#### 3.3.3. Post-Transition Intervention

This type of transition intervention occurs in adult clinics and focuses on assessing treatment outcomes after the transition has occurred. This is implemented to inform better strategies to transition ALHIV and examine if the transition was successful or not [30]. Three studies [2,23,26] incorporated this intervention alongside other types of transition interventions.

#### 3.3.4. Combination Interventions

Three studies [23,24,26] incorporated all three stages of transition intervention. 

For the first study [24], the pre-transition intervention involved complete clinical, immunological and virological evaluations. Data on disease knowledge, adherence to therapy and psychological status (including self-esteem) were gathered from adolescents who already knew their HIV status. The during-transition phase involved a joint medical visit run by a multidisciplinary team involving pediatric and adult clinic physicians every two months. The visits were partly made at the adult clinic in the presence of pediatric staff. The interventions at this stage also involved 30 to 60 min of education sessions with each patient at the pediatric clinic. Psychological support-oriented individual meetings were conducted by a psychologist monthly and group discussions that involved adolescents and their families twice a year. A nurse and a gynecologist provided information on HIV transmission prevention and ways to protect themselves in the advent of sexual activity. 

The post-transition interventions included adolescents’ clinical evaluation, HIV immunological class, CD4+ count and percentage and HIV viral load assessments. The interventions also included administering the Psychological General Well-Being (PGWB) questionnaire and psychological follow-up. 

The second study [26], piloted a transition protocol covering pre-, during and post-transition interventions in five different phases. For the pre-transition intervention, the study focused on discussions with adolescents on the need to transition to adult care and the introduction of adolescents to the adult physician at the special adolescent clinic. The during-transition intervention involved adolescents being cared for at the special adolescent clinic by the adult physician, followed by their first appointment at the adult clinic in which the same adult physician attended to them. The post-transition intervention is an evaluation process that happens one year after transition in a follow-up appointment conducted by the special clinic adolescent Social Worker and Peer Advocate. After that, their judgment is conveyed to both the special adolescent clinic team and adult infectious disease physicians to improve the process.

For the third study [23], the pre-transition intervention involved an interprofessional team providing an individualized care plan for patients one day a week, a meeting between patients/caregivers and adult providers before the transition and psychological support for patients and families. The during-transition intervention involved health and sexual education interventions for increased independent medication management and cART adherence during follow-up and education programs. The post-transition intervention involved a survey investigating the efficacy and quality of the transition to adult care and implementing further follow-up.

### 3.4. Components of Transition Interventions

Six components of transition interventions were focused on across the included studies. These are individualized care plans, group transition programs, communication, psychological support, health and sexual education as well as mHealth. 

#### 3.4.1. Individualized Care Plan

The majority (six of seven) of the transition intervention focused on providing care, support, education and counseling to individual adolescents. This was implemented in variable appointments or times to suit individuals, tailored antiretroviral treatment to meet individual needs, and individualized counseling, therapy or sessions.

#### 3.4.2. Group Transition Program

Only one transition intervention focused on transitioning adolescents as a group [29]. This involved a camp meeting that accommodates interaction between adolescents and adult clinic healthcare professionals and their peers. Afterward, the transition takes place as a group. The adolescents’ first appointment in the adult clinic is a group appointment. This is to provide peer support and proper integration during the transition.

#### 3.4.3. Communication

A steady flow of communication between patients/pediatric or adolescent healthcare givers and adult clinic specialists was emphasized in four studies [23,24,28,29]. This was noted to give adolescents a voice in the programs and enhance the efficient transitioning process between pediatric/adolescent clinic staff and the adult clinic staff.

#### 3.4.4. Psychological Support

Three studies [23,24,28] incorporated psychological support for adolescents and their families in their approach to effective transition from pediatric to adult health care. One study provided psychological support by meeting needs associated with diagnosis disclosure, HIV acceptance, the weight of living with a chronic disease and demanding family structures or dynamics [23]. The second study, on the other hand, was psychologically supported by conducting an individual meeting once a month by a psychologist knowledgeable in pediatric HIV infection management to adolescents [24]. Additionally, group meetings were conducted with all adolescents and their families twice a year. The third study) utilized 30–60 min individual sessions that allowed counselors to provide psychological support to HIV-positive youth [28].

#### 3.4.5. Health and Sexual Education

Six studies implemented health and sexual education in their transition approach [23,24,26,27,28,29]. One study provided “a 30 to 60 min face-to-face education session focusing on HIV infection, its clinical manifestations, risks, transmission routes and prevention measures and principles of ART and adherence to each patient” [24]. Another study utilized a transition camp to educate on antiretroviral management, the transmission of HIV and piloting adult HIV care settings [29]. Another study used a sexual education and risk reduction counseling approach which focused on safe sex behaviors, relationships, reproductive health, family planning and healthy conception, reducing risk behaviors for HIV and STIs and HIV-disclosure and partner testing [28]. In addition, the study provided health knowledge on transitioning to adult HIV care, self-care (e.g., ART adherence, keeping appointments) and knowledge about Health Care Insurance [28].

Another study incorporated health and sexual education by providing client education and peer counseling [26]. 

Another study focused its health and sexual education on reducing risk behaviors for HIV transmission, increasing positive lifestyle and increasing independent management of medications to increase cART adherence [23]. The last study employed the services of a sexologist who helps to deal with adolescents’ concerns about their sexuality, sex and cohabitation [27]. This approach ensures they practice safe sex and avoid HIV transmission to their partners (though it was noted that not many of them were sexually active). They were also giving education on contraceptive use.

#### 3.4.6. mHealth Intervention

One study compared transition treatment outcomes between adolescents receiving Access Care Early (ACE) transition programs and those receiving Standard of Care (SOC) [25]. In addition to the automated appointment reminders received by SOC, all ACE patients received phone or text message bi-directional communication from their peer navigator. “Bidirectional communication with the peer navigator was defined as a telephone or electronic message conversation between the patient and the peer navigator. The unidirectional conversation was defined as a telephone or electronic message left by the peer navigator without documented patient response to the message” [25]. A significant association was reported between retention in care and frequent social work visits, nurse phone calls and peer navigator interactions. More ACE patients were retained in care compared with SOC patients.

### 3.5. Primary Outcomes of Transition Interventions

All studies included in this review except one [26] reported more than one outcome of interest. Six studies reported retention in care, three reported adherence, and four reported viral load suppression.

#### 3.5.1. Retention in Care

Six out of the seven studies reported retention in care. Most of the studies [23,24,26,28,29] measured retention in care as a percentage of adolescents still in care, excluding those lost to follow-up, those who died and those who transferred or moved out. Only one [25] used a standard definition for retention measurement, which is the United States Health and Human Services Health Resources and Services Administration (HRSA) HIV/AIDS Bureau Performance Measure for HIV medical visit frequency measured as “at least one medical visit in each 6-month period of a 24-month period with a minimum of 60 days between the first medical visit in the prior 6-month period and the last medical visit in the subsequent 6-month periods”. Additionally, ACE participants included in the study have a high risk of attrition and viremia [25]. The time point for retention in care ranges from 1-year [26] to 10 years [23]. The percentage retention in care spans 49% in [25] at two years to 92% in [24] at one year and six months. 

#### 3.5.2. Adherence

Of the seven studies, three reported adherence to antiretroviral therapy [23,27,28]. The percentage of patients who adhered to treatment was 92%, 99% and 88%.

#### 3.5.3. Viral Load Suppression

Four studies [24,25,27,29] reported viral load suppression. Viral load was measured in copies/mL. Variable definitions were used to measure viral load suppression in the studies. Two studies [24,29] measured viral load suppression at <40 copies/mL. One study [27] measured it as <50 copies/mL, while another study [25] measured it at <200 copies/mL. The percentage of virologically suppressed patients ranged from 60% in [25] to 92% in [24].

### 3.6. Secondary Outcomes

Psychosocial wellbeing and self-efficacy were reported in only two [23,24] of the included studies. One study measured general psychological wellbeing using the Psychological General Well-Being (PGWB) index and self-esteem using the Multidimensional Self-Esteem Test (TMA) [24]. While psychological wellbeing increased in adolescents, there was no significant increase in self-esteem. Another study reported self-efficacy concerning patients’ ability to manage their medication independently [23]. Thirty out of 38 (79%) managed their medication without anyone’s assistance.

### 3.7. Quality Assessment of Studies

The risk of bias and quality assessment for all included studies was based on the National Institutes of Health (NIH) quality assessment tool for observational cohort and cross-sectional studies. This tool consists of 14 items. These items were scored for each included study and classified as “good”, “fair”, or “poor”. For a study to be rated “good”, it has to be positive for at least eleven of the items and for “fair”, nine of the items, while for “poor”, eight of the items and below. This classification helped to factor in different bias judgments for the included studies. Three studies [24,25,28] were rated as good quality, which is equivalent to a low risk of bias. Confounding was controlled for in these studies except for one study [24] (the only question receiving a “no” in this study). One study [26] was rated as fair quality, equivalent to a moderate risk of bias. It fulfilled nine items cut-off point for fair, which highlighted that its participation rate was less than 50% and confounding was not controlled for. The other three studies [23,27,29] were rated as poor quality equivalent to a high risk of bias. This is because they had positive eight items and below, which highlighted that exposure and/or outcome measures were not defined, sample sizes were not justified, no effect estimates and confounding variables were not adjusted statistically.

## 4. Discussion

### 4.1. Scope and Quality of Interventions

This review found few transition intervention studies that have been formally evaluated. Even though several studies described transition interventions, they did not evaluate the outcomes. In other studies, no transition interventions were implemented before the treatment outcomes were measured. Hence, they were excluded from this review.

All the included studies had sample sizes that fall within the small to moderate size range. Small sample sizes have large variations and are prone to an inaccurate estimate of the true effect size [31]. This issue could lead to the non-generalizability of the findings in other settings. In addition, reliability regarding intervention efficacy could also be hampered. 

The methodologies of the studies included in this review were cross-sectional (two), retrospective cohort (two) and prospective cohort studies (three). Since most of these studies made use of education, it was difficult to introduce RCTs.

The current review found no randomized control trials (RCTs) to evaluate the effectiveness of transition interventions for adolescents on ART on treatment outcomes. Generally, randomized control trials with adequate sample size and appropriate blinding are widely regarded as providing the highest level of evidence in epidemiological studies [32,33]. Well-designed RCTs have fewer systematic errors and biases than observational studies such as cross-sectional surveys and retrospective cohorts [34]. Randomization also helps prevent biases, minimize confounding, produce comparable intervention groups and ensure a higher level of result reliability [35]. Therefore, it is strongly recommended that future transition research involve rigorous designs (such as RCTs) to measure the effects of interventions on treatment outcomes for adolescents on ART.

All transition interventions included in the current review were conducted in healthcare facilities. No community-based transition interventions have focused on adolescents on ART. This poses a window of opportunity for HIV transmission, especially among adolescents living in underserved areas with little or no access to healthcare facilities. Differentiated care, such as community adherence groups in Mozambique, can be extended to adolescents on ART in underserved areas [36]. A community-based transition intervention can help alleviate the heightened stigma adolescents experience at this developmental stage of their lives [37]. Extending interventions beyond the health system could be a possible solution to the social determinants of health, particularly for adolescents on ART [38].

This review’s transition intervention studies involved both perinatally infected and behaviorally infected adolescents. However, most of the studies mainly focused on perinatally infected adolescents. There are concerns with using the same intervention for behaviorally and perinatally infected adolescents. Hence, some studies have recommended that behaviorally infected adolescents be researched [26]. 

### 4.2. Effectiveness of the Six Components of Transition Interventions

Transition interventions across studies are heterogeneous and differentiated care is provided [12]. Six components of these heterogeneous and differentiated care were highlighted across the included studies in this review. These are individualized care plans, group transition programs, communication, psychological support, health and sexual education and mHealth. All studies in this review use an interprofessional/multidisciplinary team, even though the teams’ professionals differ across studies. 

For *individualized care plan*—It includes considering working hours for employed adolescents, school hours for those attending schools, individual transition readiness to transfer for all adolescents, individualized counseling sessions, personalized ART treatments provided according to individual medical history and personalized psychological support [26,27,29]. This approach improved adherence, retention in care, and viral load suppression in almost all the studies at post-transition over the short and long term.

For *group transition program*—This is a program in which adolescents are transferred to the adult clinic as a group to provide peer support. One study reported that the approach provided technical support in 16 other locations [29], albeit with low-quality evidence. Hence, there is a need for rigorous evaluation designs to measure the effectiveness of group transition on treatment outcomes for adolescents on ART.

For *communication*—This points to the free flow of communication between sending and receiving health staff and between adolescents and receiving health staff. Communication interventions improved the management of each adolescent on ART at both clinics and helped them familiarize themselves with receiving health staff, while also easing the transition fears of sending clinic staff. This intervention improved adherence, retention in care, and viral load suppression post-transition over the short and long term.

For *psychological support*—This involves the expertise of psychologists knowledgeable in pediatrics HIV. The health staff meets the needs of adolescents during HIV disclosure, supporting them to accept their HIV status, helping them navigate demanding family dynamics, and providing one-on-one sessions with adolescents on ART and their family members to combat mental situations such as depression or its symptoms. This has been shown to improve self-efficacy and promote low psychologic distress among adolescents on ART, leading to better treatment outcomes. Psychological support improved adherence, retention in care, and viral load suppression post-transition over the short and long term.

For *health and sexual education*—This involves educating adolescents on HIV infection, its clinical manifestations, risks, transmission routes and prevention measures, principles and management of ART, safe sex behaviors, relationships and partner testing, reproductive health and family planning. Peer counseling can be used to educate adolescents. A sexologist can also be employed to deal with concerns about adolescents’ sexuality. This approach was found to improve adherence, retention in care and viral load suppression.

For *mHealth Intervention*—This involves using telecommunication devices for adolescents as appointment reminders. This could be in the form of phone or text message communication that is bi-directional from a peer navigator in addition to automated appointment reminders. This intervention improved retention in care and viral load suppression post-transition over the short and long term.

### 4.3. High-Income Settings Bias

There is a lack of Sub-Saharan African studies evaluating transition interventions for adolescents on ART. This is quite worrisome and concerning because most countries with the highest number of adolescents living with HIV are in Sub-Saharan Africa [3]. This paucity of research could be attributed to the lack of national and clinical guidelines and protocols in most of Sub-Saharan Africa [14]. Additionally, a study [9] reported that validated tools for assessing transition interventions among adolescents on ART are lacking in Sub-Saharan Africa. There is a need for strategic support in Sub-Saharan Africa to develop national guidelines and protocols focused on adolescents’ transition (pre-, during and post-) from pediatric/adolescent clinics to adult care. 

### 4.4. Recommendations for Clinical Practice

The included studies show that transition interventions that integrate individualized care, communication between sending and receiving staff and between adolescents and receiving staff, psychological support, health and sexual education and a mHealth approach can improve treatment outcomes among adolescents on ART. The findings of this review can be utilized by health providers involved with ALHIV care to develop and implement transition guidelines that have been proven to support positive treatment outcomes among adolescents on ART. A multidisciplinary/interpersonal team needs to be formed to develop the transition guidelines. This team will help develop and implement the transition guidelines. The transition guidelines should first focus on an individualized care plan by considering the different realities experienced by adolescents on ART. Second, the transition guidelines should provide a robust communication and client data-sharing system between the clinics and strategic familiarization and integration of clients with the adult clinic. Third, a psychological support structure should be built into the guidelines. This will particularly help identify adolescents struggling to transition and enable specific transition interventions can be provided to them according to need. Four, health and sexual education should start early pre-transition and continue throughout the transition process. Five, a mHealth approach such as appointment reminders and communication exchange between adolescents and their health providers, such as peer navigators, should be clearly stated and provided in the guidelines. Lastly, monitoring and evaluation should be incorporated into the guidelines throughout the transition process to give room for reviews and adjustments as needed.

Overall, integrating all these factors by healthcare professionals could improve ALHIV’s healthcare services and inform their better treatment outcomes both in the healthcare settings and in the community.

### 4.5. Recommendations for Further Research

This review fills a gap in the literature by describing available transition interventions, their quality of evidence and highlighting what works for transitioning adolescents on ART from pediatric/adolescent to adult clinics. This review also reveals an absence of quantitative research on this topic in Sub-Saharan African settings where adolescents on ART are predominantly domiciled and health resources are most scarce. This calls for more rigorous evaluation studies on transition interventions and particularly in the Sub-Saharan Africa region. There is also a lack of transition policies and protocols in Sub-Saharan Africa at the national and clinical levels. More concerted research efforts should be put in place to develop transition policies and protocols that are tailored to suit Sub-Saharan African countries. The available studies are poorly powered because of their small sample sizes, and there is no study using randomized control trials. Hence, better-powered studies and trials are needed to evaluate the effectiveness of transition interventions focused on adolescents on ART. These studies will generate high-level evidence that can be used to determine the effectiveness of interventions with a minimal level of bias.

## 5. Conclusions

This systematic review has described the few available transition interventions targeted at adolescents on ART, which include individualized care plans, group transition, communication between the pediatric and adult clinic and between adolescents and adult clinic staff, psychological support (clinical, family and peer), health and sexual education and mHealth. Based on the strength of evidence of the included studies, we strongly recommend transition interventions that include a combination of individualized care, communication between sending and receiving staff, psychological support, health and sexual education and a mHealth approach to improve treatment outcomes among adolescents on ART. We also recommend familiarization of adolescents on antiretroviral therapy with adult care physicians both at the sending and receiving clinics. We recommend further research with rigorous evaluation designs to measure the effects of group transition on treatment outcomes for adolescents on antiretroviral therapy. We note the lack of RCTs in this review and recommend rigorous evaluation study designs to determine the effectiveness of interventions noted with high potential for success (evidence of positive effects).

## Figures and Tables

**Figure 1 ijerph-19-14911-f001:**
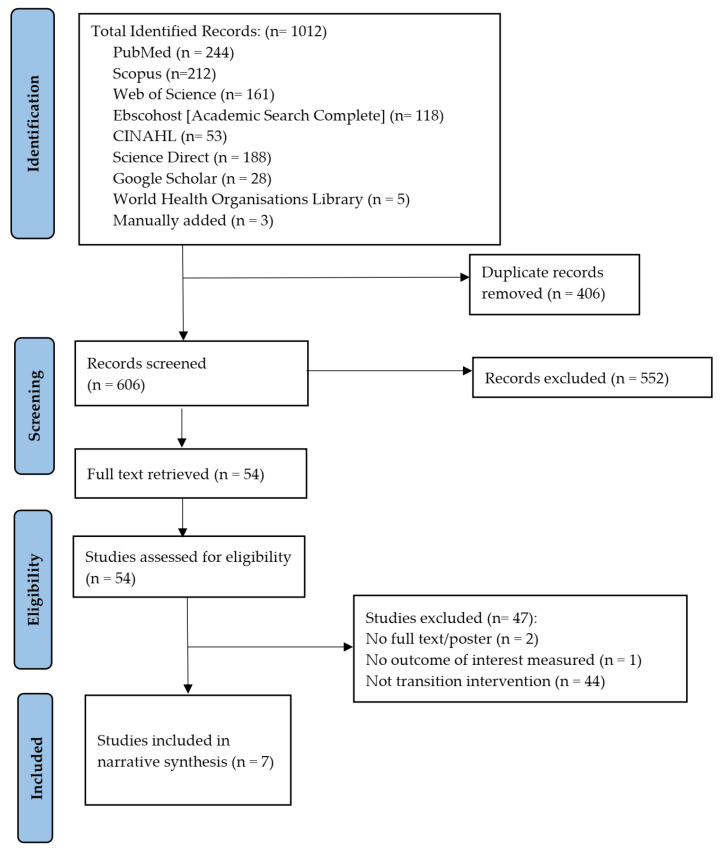
Prisma flow diagram for the review.

**Table 1 ijerph-19-14911-t001:** Picot Criteria.

**P**	Population	Adolescents Ages 10–19 Years, Living with HIV on Antiretroviral Therapy.
**I**	Intervention	Transition interventions for adolescents on HIV treatment (ART).
**C**	Comparison	Nil as observational studies will be included in this review
**O**	Outcome(s)	Primary outcomes—Adherence to ART, retention in care, or viral load suppression. Secondary outcomes: psychosocial wellbeing and self-efficacy.
**T**	Time	2000 to 2021

**Table 2 ijerph-19-14911-t002:** Five studies used a cohort study design [24,25,26,28,29] and two were cross-sectional surveys [23,27]. The sample sizes ranged from small (13) to moderate sample size (192).

First Author, Year	Country	Study Population, (Sample Size)	Study Design	Description of Intervention	Outcomes	Results
Righetti, 2015 [23]	Italy	Children and Adolescents (2–18 years) *n* = 45 [2–9 years: *n* = 25 10–18 years: *n* = 20]	Cross-sectional survey	Interprofessional team provide an individualized care plan for patients Child-customized environment A meeting between patients/caregivers and adult providers before the transition Psychological support for patients and families Health and sexual education Education for increased cART adherence (all education is part of transition intervention)	Retention in care (at 10 years)	84.4% (38/45)
					Adherence (of those retained in care)	92.1% (35/38)
					Viral load suppression (of those who adhered)	91.4 (32/35)
					In total participants <50 copies/mL	71.1% (32/45)
					***Secondary outcome*** Self-efficacy	79.0% (30/38)
Griffith, 2019 [25]	United States	Young adults (18–30 years) *n* = 137 [Accessing Care Early (ACE): *n* = 61 Standard of Care (SOC): *n* =76]	Retrospective cohort	Youth-focused care Weekly review of patients by a multidisciplinary team Appointment reminders from the peer navigator. Variable appointment times and schedules.	***Primary outcomes*** Retention in care (At 24 months)	18 transferred out or moved (10 ACE and 8 SOC) **ACE** vs. **SOC:** 49% vs. 26%, *p* < 0.001 (25/51 vs. 18/68)
					Virologic suppression (At 24 months)	For those who were retained: **ACE** vs. **SOC:** 60% (15/25) vs. 89% (16/18) (*p* = 0.04)
Maturo, 2015 [26]	United States	Adolescents and young adults (14–23 years) *n* = 38	Retrospective cohort	Preparation for transition with special adolescent clinic (SAC) team. Communication between SAC and adult care physician Familiarization of clients with the adult care physician both at the SAC and adult clinic One-year follow-up after the transition [SAC provides individual/group therapy, client education, peer counseling]	Retention in care (At 12 months)	55% (21/38)
Lolekha, 2017 [28]	Thailand	Youth (14–22 years) *n* = 192	Prospective cohort	One-day weekend workshop Two half-day meetings Three individual sessions over 18 months	Retention in care (At 18 months Retention in care (at 12 months post-intervention)	84% (161/192) 83% (134/161)
					Adherence (>95%) (At 18 months) for those retained in care	70% (159/192) 99% (159/161)
Continisio, 2020 [24]	Italy	Adolescents (13–20 years) *n* = 13	Prospective cohort	Joint pediatric/adult transition clinic running every two months Education session Psychological support—Once a month Group discussion with patients and their families twice a year Simplification of ART therapy	Retention in Care (At 18 months)	92% (12/13)
					Viral load suppression (At 18 months)	92% (12/13)
					** *Secondary outcomes:* ** Psychological general wellbeing Self-esteem	
Westling, 2014 [27]	Sweden	Young people (17–25 years) *n* = 34	Cross-sectional survey	A monthly transfer reception run with a visit every three months per patient in the evenings. care adapted to individual needs Sex and cohabitation counseling adherence to treatment education	Adherence (At six months)	88% (30/34)
					Viral load suppression (< 50 copies/mL) (At 6 months)	79% (27/34)
Hansudewechakul, 2015 [29]	Thailand	Adolescents (11–18 years) *n* = 67	Prospective cohort	A group transition program 1–2 days transition camp Communication between adolescents and adult HIV care providers Introduction of adolescents to different sections of the adult clinic Transferred as a group after camp.	Retention in care (At six years) Viral load <40 copies/mL (at 6 years)	73% (49/67) 76% (37/67)

## Data Availability

Not applicable.

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
