# Peer review of "Transition Interventions for Adolescents on Antiretroviral Therapy on Transfer from Pediatric to Adult Healthcare: A Systematic Review"

_ijerph, 2022, doi:10.3390/ijerph192214911_

Round 1

Reviewer 1 Report

This manuscript reviewed transition interventions for adolescents on antiretroviral therapy from pediatric to adult. The authors well reviewed previous clinical trials; however, the reviewer has some questions. Please respond the following questions.

1) The cited clinical researches include educations by healthcare providers. In such studies, it is difficult to introduce RCT. The authors should discuss what kinds of clinical studies are the most suitable. The reviewer thought that simple comparison of before and after introduction of education were the most suitable.

2) Considering difficulty of setting control groups, effectiveness of the transition intervention is not always proved. Please discuss it.

3) The patients who want to receive the intervention may tend to show good compliance without the intervention. Please discuss it.

Reviewer 2 Report

You have indicated that three additional studies were added manually on searching references in the included studies. Please indicate this on the PRISMA diagram. 

Reviewer 3 Report

I think it is a valuable attempt to review systematically transition interventions for adolescents on antiretroviral therapy. Although it is a bit disappointing to review at a time when many studies on this topic have not been done yet, it can be evaluated that it provides valuable information for clinical work and future studies. As a systematic review study, no major methodological problems were found, so it would be a good thesis if only meticulous revisions were made in relation to the writing of the paper. Somethings need to be supplemented in this manuscript are as follows.

1. I would like to suggest the necessity of this study in a little more detail in the introduction.

2. If the first paragraph is divided into two paragraphs, it will be easier to read.

3. Are there really only 7 studies worldwide on this subject that could be included in this study?

4. It is recommended that lines 343 through 351 be one paragraph.

5. APA style is found, not the citation style of this journal. (Line 31, 43, 44, 104, 180, 190, 197). It is recommended that the studies being reviewed should be presented in a table and presented by study number.

6. You made good suggestions for future research. However, the clinical implications need to be more specific.
